# HandsOff: Labeled Dataset Generation with No Additional Human Annotations

## Abstract

Because of their success in producing realistic images, generative adversarial networks (GANs) have recently been leveraged to generate labeled synthetic datasets. However, existing dataset generation methods do not sufficiently leverage existing images with high quality labels, which often limits either the practicality of the system or the complexity of generated labels. We propose the HandsOff framework, which is capable of producing an unlimited number of synthetic images and corresponding labels after being trained on a small of number of *pre-existing labeled images*. Our framework avoids the practical drawbacks of similar frameworks while retaining the ability to generate rich pixel-wise labels, such as segmentation masks. This capability is achieved by unifying the field of GAN inversion with synthetic dataset generation, providing a new application for GAN inversion techniques. We demonstrate the efficacy of our framework on semantic segmentation tasks by generating labeled image datasets, and training and evaluating the performance of a downstream task. Our method achieves state-of-the-art performance in synthetic data trained semantic segmentation on both the CelebAMask-HQ dataset and Car-Parts-Segmentation dataset, and produces high quality segmentations in both domains. In addition, our framework uses significantly fewer computational resources than prior work, demonstrating the supremacy of our approach in performance, annotation, and computation.

## 1 Introduction

The strong empirical performance of modern machine learning models has been enabled, in large part, by vast quantities of hand labeled data. Labeling massive datasets, such as ImageNet [1], requires a large time and cost investment. In contrast, collecting large quantities of *unlabeled* data is relatively easy. As a result, large quantities of unlabeled data exist alongside a small number of existing labeled images in many domains [1–3]

Recently, generative adversarial networks (GANs) [4–6], such as StyleGAN [7] and its variants [8–10], have demonstrated an ability to generate highly realistic images in numerous domains. Remarkably, the latent space of these networks form rich representations of images in a disentangled manner [11–13], which can be utilized to edit or remove complex semantic attributes in generated images. The ability to identify semantically meaningful parts of generated images in the latent space suggests that such representations could be used to generate pixel-level labels. This capability, coupled with GANs' ability to generate vast troves of high quality images, could serve as the basis for generating synthetic image *datasets*.

In this work, we propose the HandsOff dataset generating framework, which is capable of producing synthetic images with corresponding labels. HandsOff leverages the expressive power of the GAN latent space, the existence of high quality labeled images, and recent advances in the field of GAN

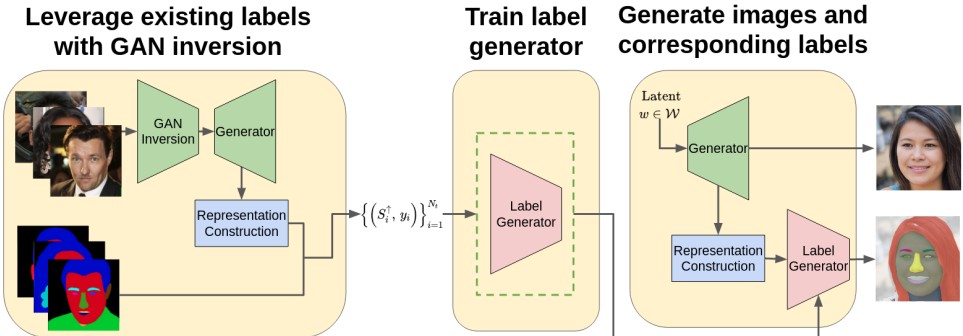

Figure 1: The HandsOff framework. GAN inversion is used to obtain training image latent codes $w_i$, which are then used to form hypercolumn representations $S_i^{\uparrow}$. The label generator is then trained with the hypercolumn representations and original labels. To generate datasets, the trained label generator is used in conjunction with a StyleGAN2 generator to produce image-label pairs.

inversion. We empirically validate the power of our framework in two domains: faces and cars. Furthermore, we explore directions for making the framework more computationally lightweight.

## 2   Related work

Our work is built on recent advances in GANs [4], which consist of a generator that synthesizes new images, and a discriminator that discerns between real and generated images. Specifically, we utilize the popular StyleGAN2 architecture [8], which synthesizes images by passing randomly sampled inputs through a series of *style blocks*. StyleGAN2 is known for its numerous latent spaces, such as the $\mathcal{Z}, \mathcal{W}, \mathcal{W}+$, and $\mathcal{S}$ spaces. See [14] for a more detailed discussion.

GAN inversion is the process of mapping a real image onto the latent space of a GAN. The myriad of inversion techniques range from encoder-based approaches [15–18], which utilize trained encoders to map images directly to the latent space, to optimization-based approaches [19, 12, 13], which directly optimize an image similarity based loss (e.g., LPIPS [20]) to obtain the latent code. Some GAN inversion methods modify aspects of the generator, such as weights or noise injection values, to increase image reconstruction quality. In our work, we exclusively use inversion methods that do not modify the generator, since the generator must remain unperturbed to generate new images from the original data distribution. We choose to invert images to the $\mathcal{W}+$ space by learning a different latent vector corresponding to each of the generator's style blocks. As noted in [14], the $\mathcal{W}+$ space is more expressive and leads to higher quality reconstructed images. We primarily utilize encoder methods to invert images, and use optimization methods to refine the encoder output in settings where finer details are not preserved.

Numerous approaches utilize GANs to generate synthetic datasets [21–27], typically in the zero-shot learning setting. We build upon DatasetGAN [28], which trains a label generator using representations of an image formed from the GAN latent code. However, a considerable drawback of this framework is that it requires *manual annotation of GAN generated images*. This is extremely burdensome, as new annotations are required for every new domain in which a user wishes to synthesize datasets. Furthermore, if the labeling paradigm changes, and the original labels cannot be directly mapped to the new labels, then additional annotations are again required. Acquiring additional labels is inconvenient, especially if labels already exist. EditGAN [29], a follow-up work to DatasetGAN, hints that such a framework is possible. However, their focus is primarily on image editing, whereas the HandsOff framework fully fleshes out the idea of unifying GAN inversion and dataset generation.

## 3   The HandsOff framework

The HandsOff framework, shown in Figure 1, builds upon the DatasetGAN framework by introducing the ability to generate labeled synthetic data from existing labeled images. We use the term "label" generically to emphasize that our approach applies broadly to label types such as segmentation masks, keypoints, or any other pixel-level label. In our experiments, we focus on generating semantic segmentation masks, a widely used and particularly resource intensive annotation to collect in bulk.

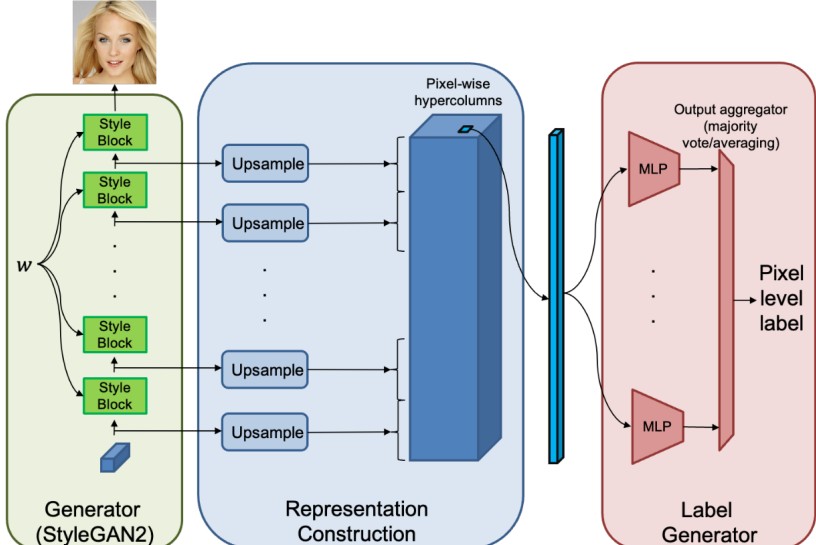

Figure 2: Construction of hypercolumns. Each intermediate style block output is upsampled to full image resolution and concatenated, resulting in a pixel-wise hypercolumn which can be fed into the label generator.

We utilize a frozen, pre-trained StyleGAN2 generator as the image generating backbone of the framework. In order to generate labels, we pass the images' latent codes through a label generator that exploits the code's unique semantic structure to efficiently generate high quality labels. Our framework crucially departs from DatasetGAN's during training. Rather than generating new latent codes and manually annotating their corresponding images to train the label generator, we instead use GAN inversion to obtain the latent codes corresponding to already labeled training images. Specifically, assume we have $N_t$ images $X_1, \ldots, X_{N_t}$ with corresponding labels $y_1, \ldots, y_{N_t}$. After passing these images through a GAN inverter, we obtain latent codes $w_1, \ldots, w_{N_t}$, which are used to form what we call the *hypercolumn representations* $S_1^\uparrow, \ldots, S_{N_t}^\uparrow$. The label generator is then trained on the $\{(S_i^\uparrow, y_i)\}_{i=1}^{N_t}$ pairs.

## 3.1 GAN inversion

The key step in the HandsOff framework is to employ GAN inversion in a new application area: dataset generation. By utilizing GAN inversion in dataset generation, we *no longer require manual annotation of GAN generated images*. Instead, a small number of pre-existing labels can be used to generate massive synthetic datasets. By using pre-existing labels, practitioners not only avoid the cost of acquiring labels, but also avoid the prerequisite of maintaining annotation workstreams in their machine learning pipelines.

One requirement of the HandsOff framework places on GAN inversion techniques is preserving the generators' original weights. This requirement ensures that the generator produces *new* images from the data distribution of the domain of interest. In our experiments, we utilize ReStyle [16], but we emphasize that our framework is amenable to *any* GAN inverter that does not modify the generator weights.

In settings where finer details are not preserved with ReStyle, we further refine the latent code obtained from ReStyle by solving a regularized optimization problem, similar to approaches in [29, 30]. Our use of optimization-based approaches is justified despite their slower inference times because GAN inversion is only performed once at the beginning on a small number of training images. In particular, let $G$ be a frozen, pre-trained StyleGAN2 generator, $X$ be the image to be inverted, and $w^{(r)}$ be the latent code obtained by running ReStyle on $X$. We refine $w^{(r)}$ by solving the following optimization problem:

$$\min_w \quad \mathcal{L}_{LPIPS}(X, G(w)) + \lambda_{\ell_2} \cdot \|X - G(w)\|_2^2 + \lambda_{reg} \cdot \|w - w^{(r)}\|_2^2. \tag{1}$$

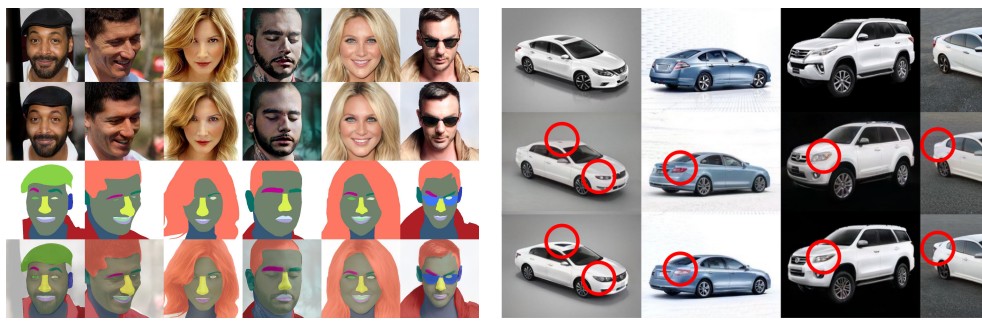

(a) Visualization of inverting real face images (row 1) with ReStyle. The reconstructed images (row 2) align well semantically with the original segmentation masks (row 3), as shown in row 4.

(b) Visualization of reconstruction quality before (row 2) and after (row 3) optimization refinement. Red circles indicate finer details that were improved to align better with the original image (row 1).

Figure 3: Visualization of image reconstruction quality for faces and car domains.

Above, $\mathcal{L}_{LPIPS}$ is the LPIPS loss [20] and $\|\cdot\|_2$ is the $\ell_2$ norm. In practice, (1) is a highly non-convex problem and using a set number of gradient descent iterations significantly refines the latent code. This refinement approach generalizes beyond ReStyle – any encoder output can be refined by (1).

## 3.2 Hypercolumn representation

Within the StyleGAN2 generator, the latent code $w$ is used to modulate convolution weights in intermediate style blocks, which progressively grow an input to the final output image. For a 1024 $\times$ 1024 resolution image, there are $L = 18$ style blocks. We take the intermediate output of these style blocks, upsample them channel-wise to the resolution of the full image, then concatentate each upsampled intermediate output channel-wise to obtain pixel-wise hypercolumns, similar to the approach in [28]. We denote the hypercolumn representation of the image as $S^\uparrow$, with each pixel $j$ now having a hypercolumn $S^\uparrow[j]$ of dimension $C$. This process is shown in Figure 2.

In practice, we cap the generated image resolution to $512 \times 512$, and downsample intermediate outputs from the $1024 \times 1024$ layers, as well as the original image. This is done for memory considerations when storing and training with these hypercolumn representations, as the dimension of each of the hypercolumns is relatively high ($C = 6080$ for $1024 \times 1024$ images). In Section 4.3, we explore using only a subset of the channels from the intermediate output in order to form the hypercolumn representation to alleviate these memory burdens.

## 3.3 Label generator

The label generator exploits the semantically rich latent space of the generator to efficiently produce high quality labels for generated images. Because the latent codes already map to semantically meaningful parts of generated images, complex vision models are not necessary to generate labels. Specifically, we utilize an ensemble of $M$ MLPs. The MLPs operate on a pixel-level, mapping a pixel's hypercolumn to a label. To generate a label for a synthetic image, we pass the hypercolumn formed by latent code $w$ through the $M$ MLPs, and aggregate the outputs to produce a lebel. Specifically, in the semantic segmentation setting with $K$ parts, the MLP performs pixel-wise classification, mapping pixel $j$'s hypercolumn $S^\uparrow[j]$ to a label $k \in \{1, \ldots, K\}$.

The $M$ MLPs are trained using a small number ($\sim$50) of pre-existing labeled images with a cross-entropy loss. In Section 4.3, we explore modifications to the label generating architecture that result in significant performance gains. We further experiment with reducing the number of MLPs in the ensemble in Appendix C.

## 3.4 Downstream task

In order to benchmark the quality of our generated datasets, we quantify the downstream performance of models trained exclusively on our datasets. After training a network on a generated dataset, we evaluate its performance on a hold out test set of real images with human annotations. We refer to this process as the *downstream task* and the trained model as the *downstream model*. Prior to training the

Table 1: Experimental results in face and car domains for semantic segmentation, reported in mIOU. Moving from 16 to 50 training images in HandsOff avoids the need to manually annotate 34 GAN generated images. We are unable to compare against DatasetGAN in the car domain because the labeling paradigm used to train DatasetGAN cannot be converted to the labeling paradigm in the Car-Part-Segmentation dataset.

|  | DatasetGAN 16 train | EditGAN 16 train | EditGAN (Encoder only) 16 train | HandsOff (Ours) |
|---|---|---|---|---|
| Faces | 0.7013 | 0.7244 | × | **0.7696** (16 train) **0.7748** (50 train) |
| Cars | N/A | 0.6023 | 0.5368 | **0.6222** (16 train) **0.6591** (50 train) |

downstream network, we follow the approach of [28, 31], and filter out the top 10% most uncertain images according to Shannon-Jensen divergence [32, 33].

## 4 Experimental results

In this section, we present qualitative results in GAN inversion, and downstream task performance in the car and face domains. For faces, we utilize segmentation masks from CelebAMask-HQ [3], and collapse the original 19 classes into 8 classes. For cars, we use the Car-Parts-Segmentation dataset [34], and collapse the original 19 classes into 12 classes. Details about class collapse are described in Appendix A.

### 4.1 GAN inversion images and original segmentation mask alignment

The underlying assumption of the HandsOff framework is that the semantic features in the reconstructed images align well with the segmentation masks. Should the reconstructed semantic features not align well, we would essentially be training the label generator with corrupted representations. Therefore, a crucial first step is verifying the fidelity of image reconstructions.

In the face domain, we utilize ReStyle for GAN inversion. As seen in Figure 3a, the reconstructed images from the latent codes obtained by ReStyle align very well with the semantic segmentation masks from CelebAMask-HQ. In the car domain, we highlight the power of our latent code refinement scheme, presented in Equation 1. ReStyle is unable to exactly preserve smaller details, such as the shape of headlights or the presence of a sunroof, as shown in the first row of Figure 3b. As a result, we refine the ReStyle output with 500 iterations of (1). As seen in the second row of Figure 3b, the optimization program is able to fine-tune the mis-aligned details, highlighted in the red circles.

### 4.2 Downstream task performance

We now discuss the performance of the HandsOff framework on the downstream task. In particular, we generate 10000 synthetic images and segmentation masks, filter out the top 10% most uncertain images, and train DeepLabV3 for 20 epochs with the 9000 remaining images. Examples of generated images and segmentation masks for the face domain can be found in Figure 4. For HandsOff, we utilize $M = 10$ MLPs in the label generating branch and report the best performing result among the combinations of network layer widths experimented with later on in Section 4.3.

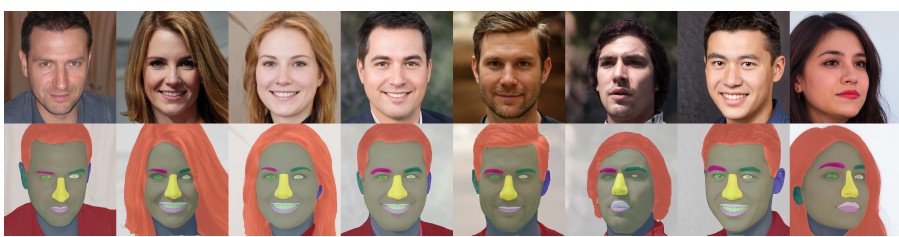

Figure 4: Examples of faces and their corresponding segmentation masks generated from the HandsOff framework trained on 50 images with *non-collapsed classes*. We do not perform class collapse here to highlight the level of detail that the label generator is able to achieve.

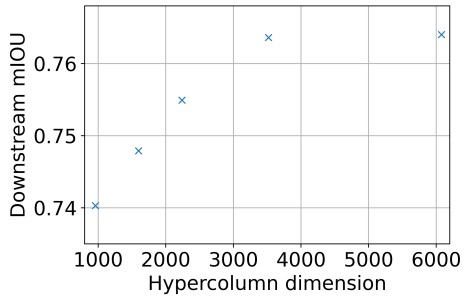

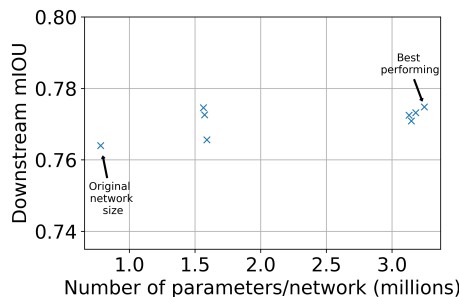

(a) Comparable performance is achieved with a 42% reduction in hypercolumn dimension. This reduction decreases the amount of memory used in training.

(b) Increasing layer widths results in relatively sizable performance gains, with the best performance occurring with intermediate layer widths of 512 and 256.

Figure 5: Framework ablations for hypercolumn dimension (left) and layer widths (right).

For faces, we split the 30000 images in CelebAMask-HQ into 3 sets: 50 images for training the label generator, 450 images for validation, and 29500 images for testing. For cars, we retain the original train (400 images) and test (100 images) splits from the Car-Parts-Segmentation dataset, using 20 images from the test set for validation. For both domains, we report mIOU of the downstream network when trained on images generated from the HandsOff, DatasetGAN, and EditGAN frameworks.

As seen in Table 1, we achieve state-of-the-art performance in synthetic data trained semantic segmentation in both the face and car domains. In particular, for faces, we highlight that increasing the number of training examples from 16 to 50 results in a sizable performance increase. Within the DatasetGAN framework, this would require manual annotation of 34 more images, a step that is not required in the HandsOff framework. Note that we are unable to test the performance of EditGAN (Encoder only) because the pre-trained EditGAN encoder weights were not publicly released.

In the car domain, the optimization based refinement of the ReStyle output results in a sizable performance increase, highlighting the importance of strong alignment of reconstructed images with the original segmentation masks. Furthermore, we are unable to run the DatasetGAN framework with the Car-Part-Segmentation labels, because the original labeling paradigm used in training the DatasetGAN framework in the car domain cannot be converted to the Car-Part-Segmentation labeling paradigm. This highlights a key drawback of DatasetGAN, as discussed in Section 2.

### 4.3 Practical modifications to HandsOff

We first experiment with keeping only a subset of the channels from the style block intermediate outputs from the lower resolution layers. In the StyleGAN2 generator, the first 10 style block outputs (which range from 4×4 to 128×128 resolutions) each contain 512 channels, comprising 5120 of the 6080 total channels. We quantify the effect of keeping zero or the first 64, 128, and 256 channels on the downstream task performance in the face domain. As shown in Figure 5a, while utilizing only higher resolution layers degrades performance considerably, we can remove 256 of the 512 channels for the first 10 style blocks with very minimal loss in performance. This results in a hypercolumn dimension 3520, which is a 42% reduction compared to the original dimension of 6080.

Finally, we investigate whether network layer widths impact downstream performance. The original DatasetGAN framework utilizes 3-layer MLPs with intermediate dimensions of 128 and 32. We explore 7 additional combinations of layer widths, as highlighted in Figure 5b. Generally, downstream performance increases with increasing network widths.

## 5    Discussion

We present the HandsOff framework, which is capable of producing high quality labeled synthetic datasets without requiring further annotation of images. We achieve state-of-the-art performance on downstream tasks in this setting, and experiment with ways of making the framework more practical from an implementation standpoint. While we focus primarily on generating semantic segmentation datasets, nothing in this framework precludes generation of continuous pixel-wise labels, such as depth maps which we plan to explore in future work.

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

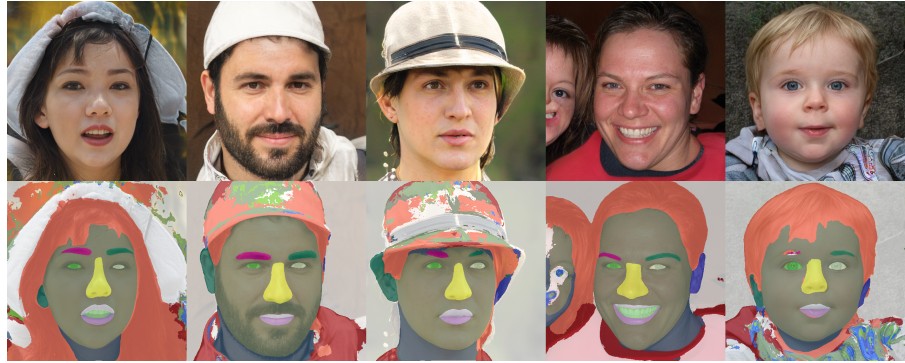

Figure 6: Examples of generated images with poor labels. These often contain components not seen in the training data (hats, multiple humans, children), suggesting that the distribution of parts in the training data has a noticeable impact on the generated labels.

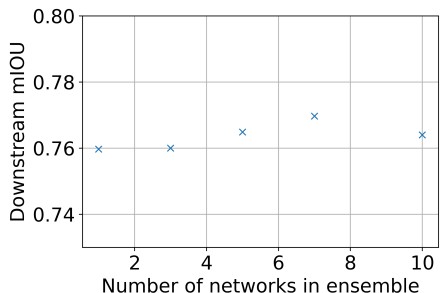

Figure 7: Downstream performance seems fairly robust to the number of MLPs in the label generator. Using fewer MLPs results in a decrease in time needed to train the framework.

## A  Dataset details

In our experiments, we collapse the original labels in each dataset in a smaller number of labeled parts. For CelebAMask-HQ dataset, we remove any distinction between left/right in a number of parts (e.g., ears, eyes, eyebrows). Furthermore, we form one mouth part consisting of upper/lower lips and mouth. Finally, we collapse all accessories and clothing into background.

For the Car-Parts-Segmentation dataset, we remove any distinction between left/right and front/back for parts such as doors, lights, bumpers, and mirrors. We also merge trunks and tailgates to be the same class.

## B  Examples of failure cases of generated images and labels

In this section, we highlight failure cases of generated labeled images. These images typically include components not included in the training data, such as humans with hats/hoods, multiple humans, and babies. Exploration of how including more edge cases in the training data affects generated label quality is an interesting direction of future work.

## C  Number of MLPs ablation

Because training an ensemble of classifiers is timely, we experiment with utilizing fewer MLPs. We train the 10 MLPs, then from the 10 trained MLPs, we use 1, 3, 5, 7, and 10 MLPs to generate labels. As seen in Figure 7, using only 1 network results in a performance drop, but using anywhere from 3 to 7 MLPs results in performance meeting or even exceeding the performance of using all 10 MLPs. This implies that there is diminishing returns in using more MLPs, in that simply using any $M > 1$ achieves (or even surpasses) the robustness benefits of using 10 MLPs.

