# OpenReview forum: "HandsOff: Labeled Dataset Generation with No Additional Human Annotations"
_NeurIPS.cc/2022/Workshop/SyntheticData4ML — Neurips 2022 SyntheticData4ML_

### Official Review · Reviewer_C74i · 2022-10-13
**Review for HandsOff**

**Rating:** 7
**Confidence:** 4

**Review:**

This paper proposes HandsOff that utilizes GAN inversion for generating synthetic data, especially, sample-part segmentation map.

HandsOff introduces a label generator that converts the feature of an image generator into labels.

For better inference of w vector, the author(s) improve ReStyle by using the reconstruction loss on LPIPS. By doing so, the reconstructed images can preserve the fine details better.

As a validation, HandsOff framework is compared with DatasetGAN and EditGAN on two datasets in terms of the part segmentation task. The experimental result demonstrates that HandsOff can be trained with fewer labeled samples than DatasetGAN and also outperform the previous works.

Considering the simple yet effective method and the outperforming performance, the paper seems to be helpful to the synthetic dataset community.

---

### Official Review · Reviewer_jk1f · 2022-10-17

**Rating:** 6
**Confidence:** 3

**Review:**

1.	Summary and contributions: HandsOff framework, capable of producing an unlimited number of synthetic images and corresponding labels after being trained on a small number of pre-existing labeled images. Labeled in this case means a segmentation mask, keypoints, or any other pixel-level label. Improvements for the DatasetGAN framework.

2.	Strengths: Usage of a pre-trained GAN network with the combination of GAN inversion and a label generator to efficiently generate labels for an image. It only needs a few images to retrain the model so it is able to produce labeled images for certain images. Modifications to the StyleGAN2 generator for higher performance.

3.	Weaknesses: The author mentions the failures of the generation of the label data, being that most of the cases are because of data not included in the training data. Is this correspondent to the data that the ReStyle network was trained or does it have to do with the data passed as a few-shot training for the network?

4.	Clarity: The ideas are clear and the results are clearly shown, due to the constraints of the paper certain discussion was pushed to the appendix.

5.	Additional feedback: MLP should be introduced as Multi Layer Perceptron (MLP). At least the failure cases of generated images should be included in the main part of the paper, these are part of the discussion and can lead to future work.

6.	Final note: Overall the paper is clear and a good addition to labeled dataset generation extending the work of the DatasetGAN framework. The limited number of pages and the total information of the paper pushed the weaknesses to the appendix leaving important parts of the discussion out.

---

### Official Review · Reviewer_yeYH · 2022-10-18
**Generating labelled synthetic data in a automated manner**

**Rating:** 4
**Confidence:** 3

**Review:**


Authors present a GAN framework to generate labelled synthetic images from a set of images.

Pros:
1. Authors address an important challenge of generating automatically labelled synthetic samples, thereby facilitating the generation of large-scale training sets. Authors employ GAN inversion to generate images and then obtain their labels.

Cons:
1. Generating labelled synthetic datasets using GANs or even autoregressive networks is not novel, and there is some work already along this line. Authors seem to have missed the entire literature. (See [1] and references therein)
2. Generating labels from images in my opinion would not preserve the semantic context and would have limited diversity. Alternatively generating the labels first and conditioning image generation on it could be more effective.
3. It is not clear how the generated labels can be trusted during test time (when using them to augment the original training data). generating labels from the images does not seem any different than just conducting semantic segmentation.

References:
[1] Cardenas et al, 2021, Generating Annotated High-Fidelity Images Containing Multiple Coherent Objects, ICIP.

---

### Meta-Review · Area_Chair_qhgf · 2022-10-19

**Recommendation:** Accept

**Review:**

Interesting problem, but not clear whether the proposed approach is better than sampling a label, then conditionally generating samples based on the label. It is also not clear how the model compares to simply generating an image and obtaining labels from a pre-trained segmentation network. I hope the authors can incorporate the reviewers comments to further improve the paper.